# Large transient assemblies of Apaf1 constitute the apoptosome in cells

Alicia C. Borgeaud [1,2], Iva Ganeva[1,2], Calvin Klein [1,3], Amandine Stooss[1], Daniela Ross-Kaschitza [1], Liyang Wu [4], Joel S. Riley [5,6,7], Stephen W. G. Tait [5,6], Thomas Lemmin [1], Thomas Kaufmann [4] & Wanda Kukulski [1,2] ✉

Upon cell death signals, the apoptotic protease-activating factor Apaf1 and cytochrome c interact to form the apoptosome complex. The apoptosome is crucial for mitochondrial apoptosis, as it activates caspases that dismantle the cell. However, the in vivo assembly mechanism and appearance of the apoptosome remain unclear. We show that upon onset of apoptosis, Apaf1 molecules accumulate into multiple foci per cell. Disassembly of the foci correlates with cell survival. Structurally, Apaf1 foci resemble organelle-sized, cloud-like assemblies. They form through specific interactions with cytochrome c, contain caspase-9, and depend on procaspase-9 expression for their formation. We propose that Apaf1 foci correspond to the apoptosome in cells. Transientness and ultrastructure of Apaf1 foci suggest that the dynamic spatiotemporal organisation of apoptosome components regulates progression of apoptosis.

Apoptosis, a major type of programmed cell death, is essential in multicellular organisms for clearance of cancerous cells and removal of excess cells during tissue development[1]. The programme relies on a cascade of molecular interactions which elicit signals that lead to cleavage of cellular components by cysteine proteases known as caspases[2]. This orchestrated dismantling of the cell results in non-inflammatory removal of the cellular remains. In intrinsic or mitochondrial apoptosis, the Bcl-2 proteins Bax and Bak accumulate on the outer mitochondrial membrane and cause its permeabilisation, resulting in the release of mitochondrial proteins into the cytosol[3–7]. Thereby, the respiratory chain component cytochrome c (cyt c) can interact with the cytosolic protein apoptotic protease-activating factor 1 (Apaf1)[8,9]. In healthy cells, Apaf1 resides in the cytosol as an inactive monomer[10–12]. Its interaction with cyt c results in conformational changes leading to oligomerisation of Apaf1 via the nucleotide-binding oligomerisation domain (NOD)[13,14]. This assembly is required for recruiting procaspase-9 which is thereby activated and processed into the initiator caspase-9[10,13,15,16]. The structure of the heterooligomeric complex formed by Apaf1, cyt c and caspase-9 is well characterised in vitro and known as the heptameric apoptosome holoenzyme[17–19]. In this wheel-like assembly, procaspase-9 binds to the central hub through interactions between its own and Apaf1 caspase recruitment domains (CARD), resulting in a local accumulation of procaspase-9 molecules, their conformational activation and self-cleavage[14,20–24].

Although the formation of the apoptosome has been elucidated in much detail using biochemistry and structural biology, the subcellular events of this pathway remain largely unknown. It is also unclear if apoptosome function is controllable after complex formation. Here, we visualised Apaf1 during apoptosis inside cells. Our results suggest that a transient, pleiomorphic assembly of Apaf1 molecules corresponds to the apoptosome and its dissociation is correlated with a lower likelihood of cell death.

## Results

### Apaf1 assembles into multiple cellular foci upon apoptosis induction

To investigate apoptosome formation by live cell imaging, we generated a HeLa cell line stably expressing Apaf1-GFP (Supplementary

[1]Institute of Biochemistry and Molecular Medicine, University of Bern, Bern, Switzerland. [2]MRC Laboratory of Molecular Biology, Cambridge, UK. [3]Graduate School for Cellular and Biomedical Sciences, University of Bern, Bern, Switzerland. [4]Institute of Pharmacology, University of Bern, Bern, Switzerland. [5]Cancer Research UK Scotland Institute, Glasgow, UK. [6]School of Cancer Sciences, College of Medical, Veterinary and Life Sciences, University of Glasgow, Glasgow, UK. [7]Institute of Developmental Immunology, Biocenter, Medical University of Innsbruck, Innsbruck, Austria. ✉e-mail: wanda.kukulski@unibe.ch

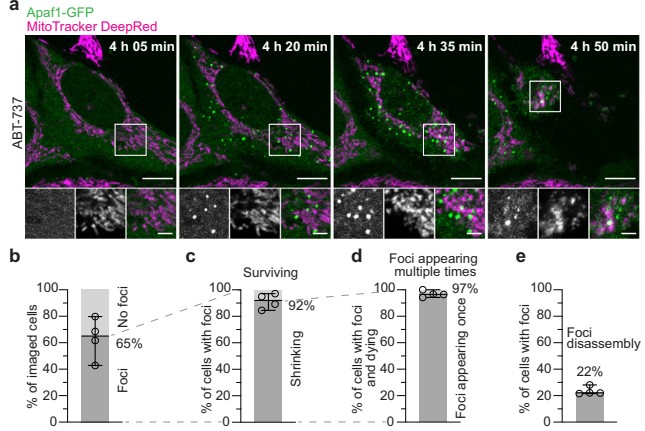

**Fig. 1 | Apaf1 accumulates into cytoplasmic foci during apoptosis. a** Live fluorescence imaging of HeLa cells stably expressing Apaf1-GFP, showing Apaf1 foci (green) upon ABT-737 treatment. Mitochondria were stained with MitoTracker DeepRed (magenta). Image acquisition time since ABT-737 treatment is indicated on large images. White squares indicate areas shown as close-ups (from left to right: Apaf1-GFP; MitoTracker DeepRed; merge). **b–e** Analysis of live fluorescence imaging. Lines correspond to medians with 95% confidence interval. $N = 327$ cells examined over 4 independent experiments. At least 54 cells were imaged per experiment. **b** Percentage of cells forming Apaf1-GFP foci, versus cells not forming foci, among all imaged cells. Median: 65%, median absolute deviation (MAD): 9%. **c** Percentage of cells shrinking (indicating cell death), versus surviving, among cells forming foci. Median: 92%, MAD: 4%. **d** Percentage of cells forming foci once, versus multiple times, among cells that shrank. Median: 97%, MAD: 1 %. **e** Percentage of cells in which foci disassembled, among all cells forming foci. Median: 22%, MAD: 0%. Scale bars in **a**: 10 µm in large images, 3 µm in close-ups.

Fig. 1). We induced apoptosis in these cells using the Bcl-2 inhibitor ABT-737[25], and imaged them by live fluorescence microscopy. We found that initially, the Apaf1-GFP signal was homogenously dispersed in the cytoplasm (Fig. 1a and Supplementary Movie 1), similarly to what has been observed before[26,27]. Over time, the Apaf1-GFP signal accumulated into dozens of foci distributed in the cytoplasm (Fig. 1a and Supplementary Movie 1). This was accompanied by fragmentation of mitochondria, typical for intrinsic apoptosis[28]. Within 18 h from induction, Apaf1-GFP foci had appeared in 65% of the cells (Fig. 1b and Supplementary Fig. 2a). The majority of these cells shrank (Fig. 1c), often followed by detachment from the imaging plate, within 23 min of foci appearance (Supplementary Fig. 2b), in accordance with progression of apoptosis (Supplementary Fig. 2c, d). Cells that did not form foci had slightly higher levels of GFP fluorescence compared to the cells with foci prior to cell death (Supplementary Fig. 2e), indicating that foci formation was not caused by excessive Apaf1-GFP levels. Taken together, Apaf1-GFP molecules assemble into areas of high concentration during the progression of apoptosis.

In a small fraction of cells, foci formation was not followed by cell death. Instead, the foci dissociated, and the cells survived (Fig. 1c and Supplementary Movie 2). In total we observed disassembly of foci in 22% of the cells that had foci, which includes cells that died only upon recurring foci appearance (Fig. 1d, e). Thus, when foci appearance was not immediately followed by cell death, disassembly of the foci could be observed, revealing their transientness. Furthermore, cells that survived showed fewer foci per imaged cell area than cells that died (Supplementary Fig. 2f), indicating that the density of Apaf1 foci per cell correlates with the probability to undergo cell death. To further investigate the transientness of Apaf1 foci, we imaged Apaf1-GFP expressing cells treated with both ABT-737 and QVD (Fig. 2a and Supplementary Movie 3). QVD is a broad-spectrum caspase inhibitor which promotes cell survival, thereby facilitating the study of transient steps[29]. Under these conditions, 41% of cells displayed Apaf1 foci

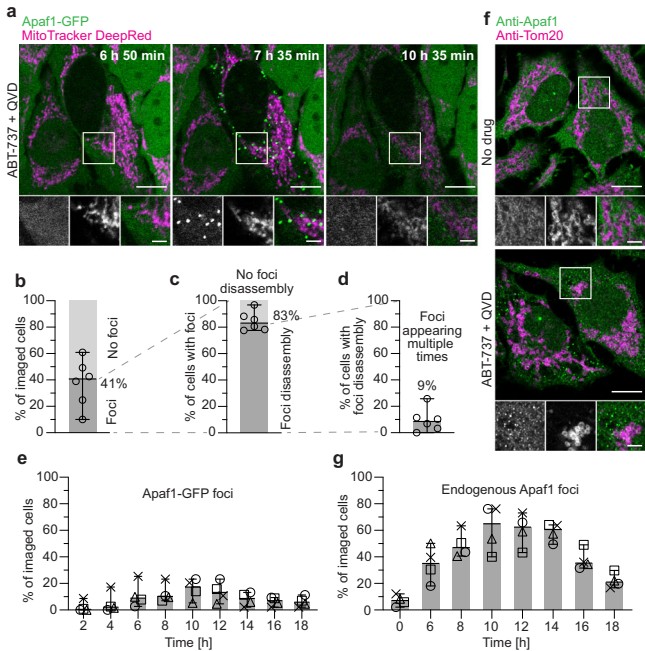

**Fig. 2 | Apaf1 foci disassemble when cells survive. a** Live fluorescence imaging of HeLa cells stably expressing Apaf1-GFP, showing assembly and disassembly of Apaf1 foci (green) upon ABT-737 and QVD treatment. Mitochondria were stained with MitoTracker DeepRed (magenta). Image acquisition time since ABT-737 and QVD treatment is indicated on large images. White squares indicate areas shown as close-ups (from left to right: Apaf1-GFP; MitoTracker DeepRed; merge). **b–e** Analysis of live fluorescence imaging. Lines correspond to medians with 95% confidence interval. $N = 942$ cells examined over 6 independent experiment for **b–d**, $N = 628$ cells examined over 4 independent experiments for **e** (see Methods). At least 94 cells were imaged per experiment. **b** Percentage of cells forming Apaf1-GFP foci, versus not forming foci, among all imaged cells. Median: 41%, MAD: 12%. **c** Percentage of cells in which foci disassembled, versus cells in which foci did not disassemble, among cells forming foci. Median: 83%, MAD: 5%. **d** Percentage of cells forming Apaf1-GFP foci repeatedly, among cells showing Apaf1 foci disassembly. Median: 9%, MAD: 4%. **e** Percentages of cells displaying Apaf1 foci at the time points after ABT-737 and QVD treatment indicated on x-axis. The different symbols represent individual experiments. At 10 h: Median: 17%, MAD: 5%. **f** Immunofluorescence of untreated as well as ABT-737 and QVD treated HeLa cells, using antibodies labelling endogenous Apaf1 (green) and the mitochondrial protein Tom20 (magenta). White squares indicate areas shown as close-ups (from left to right: Apaf1; Tom20; merge). **g** Percentages of HeLa cells displaying endogenous Apaf1 foci by immunofluorescence, fixed at the time points after ABT-737 and QVD treatment indicated on x-axis. The different symbols represent individual experiments. At 10 h: Mean: 65%, MAD: 11%, $N = 795$ cells examined over 4 independent experiments. At least 164 cells were imaged per time point and experiment. Scale bars in **a** and **f**: 10 µm in large images, 3 µm in close-ups.

(Fig. 2b) within 18 h of apoptosis induction. In most of these cells, the foci disassembled after 2 h (Fig. 2c and Supplementary Fig. 2g–i). Moreover, 9% of the cells in which foci had disassembled showed further foci appearances over time (Fig. 2d and Supplementary Movie 4). Within the imaged cell population, the percentage of cells with foci peaked 10 h after induction and decreased subsequently (Fig. 2e), reflecting the foci's limited lifetime. These results show that Apaf1-GFP foci are transient structures, and their evanescence correlates with cell survival.

Next, we tested if endogenous, untagged Apaf1 also accumulates into foci. By immunofluorescence of HeLa cells treated with ABT-737 and QVD, we observed that endogenous Apaf1 formed multiple bright foci, similarly to Apaf1-GFP (Fig. 2f). Cells displaying endogenous Apaf1 foci also had fragmented mitochondria[28]. Further, we performed time-course immunofluorescence of these HeLa cells after apoptosis

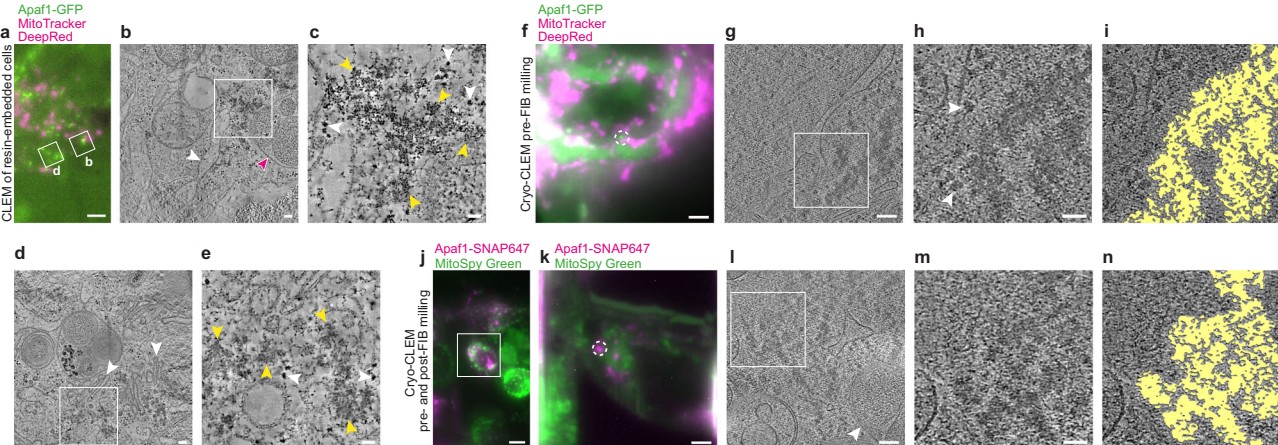

**Fig. 3 | CLEM reveals cloud-like meshwork structure of Apaf1 foci.** See Supplementary Fig. 4 for workflow details. **a–e** CLEM of resin-embedded HeLa cells transiently expressing Apaf1-GFP, treated with ABT-737 and QVD. Representative of a total of 13 electron tomograms. **a** Fluorescence image of a section through a resin-embedded cell; Apaf1-GFP (green) and MitoTracker DeepRed (magenta). White squares indicate areas imaged by electron tomography and shown in **b** and **d**. **b** Virtual slice from electron tomogram acquired at area indicated in **a**. White square indicates area that contains an Apaf1-GFP localisation, shown in **c**. Magenta arrow indicates mitochondrion, white arrow indicates endoplasmic reticulum. **c** Zoomed-in area from the electron tomogram shown in **b**. Here shown virtual slice corresponds to a different z-position than the virtual slice shown in **b**. Yellow arrows indicate parts of the cloud-like meshwork structure, white arrows indicate putative ribosomes. **d** Virtual slice from electron tomogram acquired at area indicated in **a**. White square indicates area that contains an Apaf1-GFP localisation, shown in **e**. White arrows indicate Golgi cisternae. **e** Zoomed-in area from the electron tomogram shown in **d**. Here shown virtual slice corresponds to a different z-position than the virtual slice shown in **d**. Yellow arrows indicate parts of the cloud-like meshwork structure, white arrows indicate putative ribosomes. **f–i** Pre-FIB milling cryo-CLEM of vitrified HeLa cells stably expressing Apaf1-GFP, treated with ABT-737 and QVD. Representative of a total of 3 cryo-electron tomograms. **f** Cryo-fluorescence image of a cell grown on an EM grid, imaged before cryo-FIB milling; Apaf1-GFP (green) and MitoTracker DeepRed (magenta). The white dashed circle indicates the target Apaf1-GFP focus. **g** Virtual slice from cryo-electron tomogram acquired in the target area indicated in **f**. White square indicates area shown in **h** and **i**. **h** Zoomed-in area from the cryo-electron tomogram shown in **g**. Here shown virtual slice corresponds to a different z-position than the virtual slice shown in **g**. White arrows indicate putative ribosomes. **i** Segmentation model indicating area shown in **h** that contains meshwork structure. **j–n** Pre- and post-FIB milling cryo-CLEM of vitrified HeLa cells transiently expressing Apaf1-SNAP labelled with SNAP-Cell 647-SiR (Apaf1-SNAP647), treated with ABT-737 and QVD. Representative of a total of 8 cryo-electron tomograms. **j** Cryo-fluorescence image of a cell grown on an EM grid, imaged before cryo-FIB milling; Apaf1-SNAP647 (magenta) and MitoSpy Green (green). White square indicates area targeted by cryo-FIB milling. **k** Cryo-fluorescence image of the cell area indicated in **j**, imaged after cryo-FIB milling. The white dashed circle indicates the Apaf1-SNAP647 (magenta) focus that was imaged by cryo-ET in **l**. **l** Virtual slice from cryo-electron tomogram acquired in the area indicated in **k**. White square indicates area shown in **m** and **n**. White arrow indicates mitochondrion. **m** Zoomed-in area from the cryo-electron tomogram shown in **l**. Here shown virtual slice corresponds to a different z-position than the virtual slice shown in **l**. **n** Segmentation model indicating area shown in **m** that contains meshwork structure. Scale bars in **a**: 2 µm. In **b**, **d**, **g** and **l**: 100 nm. In **c**, **e**, **h** and **m**: 50 nm. In **f** and **k**: 3 µm. In **j**: 10 µm.

induction (Fig. 2g). The fraction of cells displaying foci of endogenous Apaf1 was highest 10 h after ABT-737 treatment, and then decreased over time, in agreement with Apaf1-GFP in live cells (Fig. 2e). These experiments confirm that both formation and disassembly of Apaf1 foci are endogenous features of apoptosis in HeLa cells.

To explore whether Apaf1 foci formation also occurred in other cell types, we expressed Apaf1-GFP in U2OS and HCT116 cells (Supplementary Fig. 3a). In both cell types, we observed the formation of Apaf1-GFP foci after induction of apoptosis with ABT-737 similarly to HeLa cells. To test whether Apaf1 foci form upon induction of apoptosis with different drugs, we treated Apaf1-GFP expressing HeLa cells with cisplatin and observed that foci formed similarly to when ABT-737 was used (Supplementary Fig. 3b, c). These results indicate that Apaf1 foci formation is likely a universal phenomenon of apoptotic cells, independent of cell type and means of apoptosis induction.

## Apaf1 foci consist of cloud-like assemblies of irregular shape

Since in vitro Apaf1 oligomerises into the wheel-like heptameric apoptosome complex[10,17,19], we hypothesised that Apaf1 foci might correspond to apoptosome clusters. Therefore, we visualised the 3D architecture of Apaf1-GFP foci in HeLa cells using a correlative light and electron microscopy (CLEM) approach for resin-embedded cells, which allows to precisely correlate fluorescent signals to cellular ultrastructure[30] (Fig. 3a–e and Supplementary Fig. 4a). MitoTracker staining helped confirming the correlation based on the positions of mitochondria. We acquired 13 electron tomograms at locations that contained a total of 27 Apaf1-GFP foci in two cells. In all cases, the areas

corresponding to Apaf1-GFP foci contained cloud-like irregular meshwork structures (Fig. 3c, e). This meshwork consisted of areas with varying density, which were loosely associated with each other within one Apaf1-GFP focus and interspersed with cytosolic components such as ribosomes. While areas of high density varied in size and shape, the meshwork appeared continuous within each tomographic volume. The meshwork was found in varying proximity to mitochondria, endoplasmic reticulum or Golgi membranes, indicating no preferential localisation near a specific organelle (Fig. 3b, d and Supplementary Movies 5, 6).

While CLEM on resin-sections reliably established that Apaf1-GFP signals correlate to cloud-like meshworks, the interpretability of the identified structures might be impaired by the resin embedding. We therefore turned to cryo-electron tomography (cryo-ET), a method for imaging cellular components in a near-native state[31,32], which we combined with two different cryo-CLEM approaches (Supplementary Fig. 4b, c)[33,34]. We thereby targeted Apaf1 foci in HeLa cells expressing Apaf1-GFP (Fig. 3f–i) or Apaf1-SNAP (Fig. 3j–n). The latter showed a similar behaviour in live cells to Apaf1-GFP (Supplementary Movie 7). In the cryo-ET data obtained from both CLEM approaches we observed structures similar in appearance to those visualised in resin-embedded cells by CLEM, validating our observations (Fig. 3g, l, Supplementary Movies 8 and 9). The cloud-like, continuous meshwork associated with Apaf1 foci displayed irregular overall shapes with uneven edges (Fig. 3h, i, m, n). The meshwork structure appeared to correspond to dense, interconnected macromolecular assemblies. Within these assemblies we did not discern discrete structures that would be

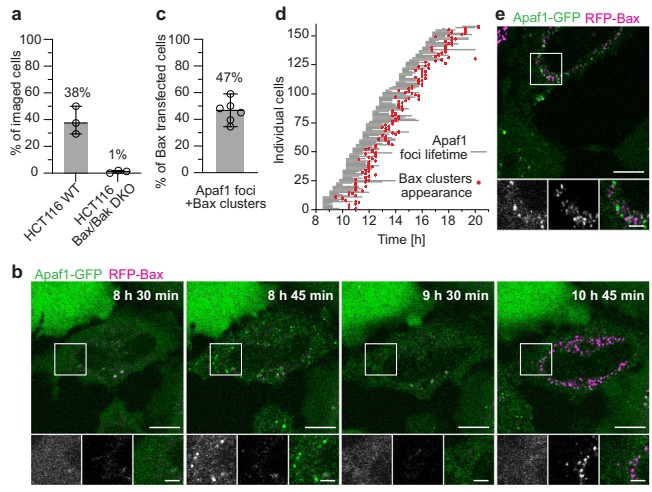

**Fig. 4 | Bax/Bak are required for Apaf1 foci formation. a** Percentage of HCT116 wild type (WT) and Bax/Bak double knockout (DKO) cells that form Apaf1 foci upon ABT-737 treatment, among all imaged cells. Lines correspond to medians with 95% confidence interval. WT: Median: 38%, MAD: 8%. Bax/Bak DKO: Median: 1%, MAD: 1%, $N = 196$ (WT) and 223 (Bax/Bak DKO) cells examined over 3 independent experiments each. At least 48 cells were imaged per condition and experiment. **b** Live fluorescence imaging of HeLa cells stably expressing Apaf1-GFP (green) and transiently expressing RFP-Bax (magenta) showing Apaf1 foci assembly and disassembly followed by Bax cluster formation. Apoptosis was induced by RFP-Bax overexpression and cells were treated with QVD. Image acquisition time since RFP-Bax transfection is indicated on large images. White squares in the large images indicate areas shown as close-ups (from left to right: Apaf1-GFP; RFP-Bax; merge). **c** Percentage of cells forming both Apaf1 foci and Bax clusters, among all RFP-Bax transfected cells. Lines correspond to median with 95% confidence interval. Median: 47%, MAD: 5%, $N = 415$ cells examined over 6 independent experiments. At least 55 cells were imaged per experiment. **d** Timing of assembly and disassembly of Apaf1 foci and Bax cluster formation per individual cell, imaged as shown in **b**. Grey lines indicate Apaf1-GFP foci lifetimes; red dots time points of RFP-Bax cluster appearance. $N = 158$ cells examined over 6 independent experiments (only cells in which Apaf1 foci disassembled during imaging are displayed). Median time between Apaf1 foci and Bax cluster appearances: 1 h 45 min, MAD: 15 min, $N = 188$ cells examined over 6 independent experiments (includes cells in which Apaf1 foci did not disassemble). **e** Live fluorescence imaging of HeLa cells stably expressing Apaf1-GFP (green) and transiently expressing RFP-Bax (magenta) in a rare case when Apaf1 foci and Bax clusters are simultaneously observable in a cell. White square in the large image indicates area shown as close-ups (from left to right: Apaf1-GFP; RFP-Bax; merge). Scale bars in **b** and **e**: 10 μm in large images, 3 μm in close-ups.

indicative of a gathering of individual apoptosome wheels. Collectively, our CLEM data indicate that Apaf1 foci consist of a higher-order assembly with a pleiomorphic ultrastructure.

### Formation of Apaf1 foci depends on Bax/Bak activity
Although the structure of Apaf1 foci does not resemble discrete protein complexes, Apaf1 foci might be equivalent to the apoptosome. To test this hypothesis, we asked whether Apaf1 foci form through a similar mechanism as the apoptosome complex. In vitro, apoptosome formation requires binding of cyt c to Apaf1[9,10]. Since the release of cyt c from mitochondria depends on permeabilisation of the outer mitochondrial membrane by Bax or Bak[4], we expressed Apaf1-GFP in an established Bax/Bak double knockout (KO) HCT116 cell line[35] and wild type HCT116 cells. We found that in absence of Bax and Bak, almost no cells displayed Apaf1-GFP foci upon induction of apoptosis, compared to 38% of wild type cells (Fig. 4a). Thus, the formation of Apaf1 foci depends on Bax/Bak activity.

We next asked which stage of Bax/Bak activity drives Apaf1 foci formation. Bax/Bak initially accumulate on the outer mitochondrial

membrane to perforate it for the release of cyt c[6], and eventually form large cytosolic clusters[36]. To assess which of these phases is relevant for Apaf1 foci assembly, we overexpressed RFP-Bax in the Apaf1-GFP HeLa cells. Bax overexpression directly induces apoptosis and allows studying Bax cluster formation[33,37]. Upon transfection with RFP-Bax, 47% of cells imaged over the course of 20 h 15 min formed Apaf1-GFP foci (Fig. 4b, c). This observation confirms dependence of Apaf1 foci formation on Bax and shows that foci formation is induced also by other means than drug treatment. Interestingly, Apaf1-GFP foci appeared 1 h 45 min before RFP-Bax clusters could be detected (Fig. 4d). As the cells were QVD treated, the majority of Apaf1-GFP foci disassembled within this time frame, in most cases before RFP-Bax clusters were detectable. In the rare cases when Bax clusters and Apaf1 foci were present in a cell at the same time, the RFP-Bax and Apaf1-GFP signals did not colocalise (Fig. 4e). In conclusion, Apaf1 foci and Bax cluster formation are temporally and spatially distinct events. Apaf1 foci formation is dependent on early Bax activity and precedes cytosolic Bax cluster formation.

### Apaf1 foci form through specific interactions with cyt c
We next wanted to directly assess whether Apaf1 foci were induced by the release of cyt c. We first microinjected bovine cyt c into Apaf1-GFP expressing HeLa cells treated with QVD[38,39]. Importantly, we did not induce apoptosis in these cells. Cyt c was co-injected with fluorescent dextran to track microinjected cells (Fig. 5a). As a control, cells were microinjected only with fluorescent dextran. Cyt c microinjection promoted Apaf1 foci formation in 35% of the microinjected cells, while no Apaf1 foci were observed in cells microinjected with dextran only (Fig. 5b, Supplementary Movie 10). The foci were transient, similarly to those induced by ABT-737 in presence of QVD (Fig. 2a, c). In summary, the availability of cyt c in the cytosol is sufficient to trigger Apaf1 foci formation.

In vitro, cyt c is known to bind to the Apaf1-WD40 repeats, leading to apoptosome formation[10,14,17–19,23,40]. To understand the molecular interactions underlying Apaf1 foci formation in cells, we generated a mutant Apaf1 expected to be unable to bind cyt c and consequently to remain inactive, which we refer to as Apaf1-dead-GFP. Specifically, we mutated Trp884 and Trp1179 in the WD40 repeats to aspartates. These two tryptophans were described as crucial for binding of cyt c to Apaf1 in the apoptosome complex[19]. We transiently expressed Apaf1-dead-GFP in HeLa cells subsequently treated with ABT-737 (Supplementary Fig. 1a). Only 5% of the cells expressing Apaf1-dead-GFP formed foci, compared to 35% of cells transiently expressing wild type Apaf1-GFP (Fig. 5c and Supplementary Fig. 5a). Thus, foci formation by the Apaf1-dead-GFP mutant is impaired. The low percentage of cells that displayed Apaf1-dead-GFP foci might arise from the presence of endogenous Apaf1 in these cells (Supplementary Fig. 1a), possibly promoting formation of heterotypic foci and cell death (Supplementary Fig. 5b). Nevertheless, these experiments show that the specific interface between cyt c and the Apaf1-WD40 repeats[19] is critical for formation of Apaf1 foci.

While these results are in line with the expectation that cyt c and Apaf1 would directly interact[9,10], we did not observe enrichment of cyt c in the Apaf1 foci by immunofluorescence (Supplementary Fig. 5c). This could be due to epitope inaccessibility of Apaf1-bound cyt c. However, it could also reflect a transient interaction between Apaf1 and cyt c, as previously suggested[27,41]. To test this possibility, we microinjected individual Apaf1-GFP expressing cells twice with cyt c, at an interval of 10 h. We reasoned that if doubly microinjected cells formed Apaf1 foci only once, it would mean that Apaf1 is not reactivatable by fresh cyt c, despite foci disassembly. Such a result could support a stable interaction between cyt c and Apaf1. However, we observed that 39% of doubly microinjected cells formed foci twice (Fig. 5d), hence after each microinjection event, suggesting that foci formation does not irreversibly affect Apaf1. This result is in line with

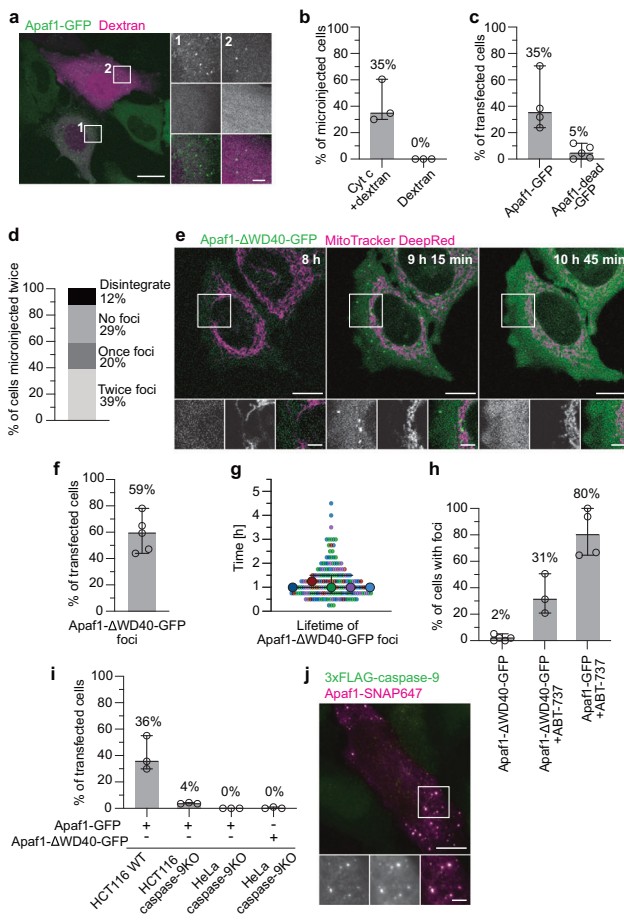

**Fig. 5 | Apaf1 foci formation depends on specific interactions with cytochrome c and on caspase-9. a** Live fluorescence imaging of HeLa cells stably expressing Apaf1-GFP (green) microinjected with bovine cyt c and fluorescent dextran (magenta). White squares indicate two areas shown as close-ups (from top to bottom: Apaf1-GFP; dextran; merge). **b** Percentages of cells forming Apaf1 foci upon microinjection of cyt c and dextran, or dextran only. Lines correspond to medians with 95% confidence interval. Cyt c + dextran: Median: 35%, MAD: 5%. Dextran: Median: 0%, MAD: 0%. $N = 81$ (cyt c + dextran) and 92 (dextran) cells examined over 3 independent experiments each. At least 16 cells were microinjected and imaged per condition and experiment. **c** Percentages of ABT-737-treated HeLa cells transiently expressing Apaf1-GFP or Apaf1-dead-GFP showing Apaf1 foci formation. Lines correspond to medians with 95% confidence interval. Apaf1-GFP: Median: 35%, MAD: 7%, $N = 592$ cells examined over 4 independent experiments. Apaf1-dead-GFP: Mean: 5%, MAD: 4%, $N = 446$ cells examined over 5 independent experiments. At least 66 cells were imaged per condition and experiment. **d** Percentages of HeLa cells stably expressing Apaf1-GFP, microinjected with cyt c twice within 10 h, that formed Apaf1 foci twice, once, never, or that disintegrated upon double microinjection. N = 41 cells examined over 1 experiment. **e** Live fluorescence imaging of HeLa cells showing foci assembly and disassembly upon transient expression of Apaf1-ΔWD40-GFP (green). Mitochondria were stained with MitoTracker DeepRed (magenta). Image acquisition time after transfection is indicated on large images. White squares in the large images indicate areas shown as close-ups (from left to right: Apaf1-ΔWD40-GFP; MitoTracker DeepRed; merge). **f** Percentage of cells that form Apaf1-ΔWD40-GFP foci during imaging. Lines correspond to median with 95% confidence interval. Median: 59%, MAD: 12%, $N = 453$ cells examined over 5 independent experiments. At least 74 cells were imaged per experiment. **g** Lifetimes of Apaf1-ΔWD40-GFP foci. Each point represents an individual cell, each colour represents an experiment. Median lifetimes of each experiment are indicated by large dots. Black lines indicate the median with 95% confidence interval of all data points combined. Median: 60 min, MAD: 15 min, $N = 189$ cells examined over 5 independent experiments. **h** Percentages of HeLa cells transiently expressing Apaf1-ΔWD40-GFP or Apaf1-GFP, in presence or absence of ABT-737, that shrank (indicating cell death) among all cells showing foci over 20 h 45 min of imaging. Lines correspond to medians with 95% confidence interval. Apaf1-ΔWD40-GFP: Median: 2%, MAD: 2%, $N = 271$ cells examined over 5 independent experiments. Apaf1-ΔWD40-GFP with ABT-737: Median: 31%, MAD: 10%, $N = 276$ cells examined over 3 independent experiments. Apaf1-GFP with ABT-737: Median: 80%, MAD: 15%, $N = 234$ cells examined over 4 independent experiments. At least 23 cells were imaged per condition and experiment. **i** Percentage of cells forming foci among all transfected cells. HCT116 wild type, HCT116 caspase-9KO, or HeLa caspase-9KO cells transiently expressing Apaf1-GFP or Apaf1-ΔWD40-GFP, as indicated. Cells expressing Apaf1-GFP were treated with ABT-737. Lines correspond to medians with 95% confidence interval. $N = 406$ (HCT116 WT + Apaf1-GFP), 391 (HCT116 caspase-9KO + Apaf1-GFP), 148 (HeLa caspase-9KO + Apaf1-GFP), 318 (HeLa caspase-9KO + Apaf1-ΔWD40-GFP) cells examined over 3 experiments for each condition. HCT116 WT + Apaf1-GFP: Median: 36%, MAD: 6%. HCT116 caspase-9KO + Apaf1-GFP: Median: 4% MAD: 0%. HeLa caspase-9KO + Apaf1-GFP: Median: 0%, MAD: 0%. HeLa caspase-9KO + Apaf1-ΔWD40-GFP: Median: 0%, MAD: 0%. At least 30 cells were imaged per condition and experiment. **j** Immunofluorescence of ABT-737 and QVD treated HeLa caspase-9KO cells co-expressing Apaf1-SNAP, labelled with SNAP-Cell 647-SiR (magenta), and 3xFLAG-caspase-9, labelled with an anti-FLAG antibody (green). White square in the large image indicates area shown as close-ups (from left to right: 3xFLAG-caspase-9; Apaf1-SNAP647; merge). Scale bars in **a**, **e** and **j**: 10 μm in large images, 3 μm in close-ups.

our observation that cells treated with ABT-737 occasionally showed repeated Apaf1 foci appearances during imaging (Figs. 1d, 2d and Supplementary Movie 4). Altogether, these results are compatible with a transient activation of Apaf1 by the specific interaction with cyt c.

We next expressed an Apaf1 mutant known to be constitutively active, as it forms the apoptosome complex and activates caspase-9 in absence of cyt c in vitro[13,42–45]. This construct lacks the entire WD40 repeats and is referred to as the Apaf1-ΔWD40 mutant[13,42] (Supplementary Fig. 1a). The WD40 repeats of Apaf1 were shown to inhibit Apaf1 activation and oligomerisation, unless cyt c is bound[43,45,46]. Thus, if Apaf1 foci form through similar mechanisms as the apoptosome, Apaf1-ΔWD40-GFP should form foci immediately upon expression, without requiring cyt c through induction of apoptosis. Indeed, 59% of HeLa cells transfected with Apaf1-ΔWD40-GFP formed foci within 45 min after GFP fluorescence was detectable in the cytoplasm (Fig. 5e, f, Supplementary Movie 11 and Supplementary Fig. 5d, e). In contrast, no cells expressing full-length Apaf1-GFP displayed foci in absence of apoptosis induction (Supplementary Fig. 5d). Thus, the lack of the WD40 repeats promotes Apaf1 foci formation in absence of apoptotic stimuli. Furthermore, for foci assembly, the N-terminal part of Apaf1 is sufficient, consisting of the CARD and the NOD domains known to drive oligomerisation into the apoptosome complex[18,19,42]. Together with our results on cyt c microinjection and the Apaf1-dead-GFP mutant, these data suggest that Apaf1 foci form through molecular interactions similar to in vitro formation of the apoptosome complex.

## Caspase-9 is required for Apaf1 foci formation and accumulates in foci

Intriguingly, the majority of foci formed by Apaf1-ΔWD40-GFP in absence of apoptotic stimuli were transient, with a lifetime of 1 h (Fig. 5g, Supplementary Fig. 5f, g and Supplementary Movie 11). Thus,

despite Apaf1-ΔWD40-GFP being spontaneously active regarding foci formation (Fig. 5e) similarly to apoptosome formation[13,42–45], this activity was evanescent. This means that foci disassembly does not require Apaf1 to go back to its autoinhibited state, since the Apaf1-ΔWD40 mutant cannot adopt it[45]. Furthermore, the ability to both assemble and disassemble foci lies in the N-terminal part of Apaf1.

We also observed that very few cells with Apaf1-ΔWD40-GFP foci proceeded to cell death despite absence of QVD (Fig. 5h). When we induced apoptosis with ABT-737, 31% of cells with Apaf1-ΔWD40-GFP foci died (Fig. 5h), probably due to the presence of endogenous Apaf1 in these cells (Supplementary Fig. 1a). In contrast, most cells with foci of transiently expressed full-length Apaf1-GFP died upon induction of apoptosis (Fig. 5h). In line with this, we observed caspase activity when

full-length Apaf1-GFP was expressed and apoptosis induced, but not when Apaf1-ΔWD40-GFP was expressed (Supplementary Fig. 5h). Thus, the constitutive formation of foci by Apaf1-ΔWD40-GFP does not promote cell death, in agreement with the behaviour of this mutant Apaf1 in the apoptosome[43,44].

In the apoptosome complex, procaspase-9 binds to Apaf1 via CARD-CARD interactions[14,23,24]. Given the ability of Apaf1-ΔWD40-GFP to form foci while consisting only of the N-terminal NOD and CARD domains, we wondered if caspase-9 plays a role in the formation of Apaf1 foci. We first expressed full-length Apaf1-GFP in HCT116 wild type and caspase-9KO cells (Supplementary Fig. 5i), and induced apoptosis with ABT-737. Only 4% of the caspase-9KO cells displayed Apaf1-GFP foci, compared to 36% of wild type cells (Fig. 5i). We next expressed Apaf1-ΔWD40-GFP and full-length Apaf1-GFP in HeLa caspase-9KO cells[47], the latter treated with ABT-737. In both cases, the transfected cells showed no foci (Fig. 5i). Thus, in absence of caspase-9, foci formation is impaired for both Apaf1-ΔWD40-GFP and full-length Apaf1-GFP. These results suggest that Apaf1 foci formation depends on caspase-9, and this dependence pertains to the N-terminal part of Apaf1 which includes the CARD domain.

Since caspase-9 is required for formation of Apaf1 foci, we asked if the foci contain caspase-9. We therefore co-expressed a 3xFLAG-tagged caspase-9 construct and Apaf1-SNAP in HeLa caspase-9KO cells treated with ABT-737 and QVD. By immunofluorescence of these cells, we found that caspase-9 colocalised with Apaf1 in foci (Fig. 5j). Foci were rare to observe when caspase-9 and Apaf1 constructs were co-expressed (Supplementary Fig. 5j), in line with an acceleration of apoptotic events upon overexpression of caspase-9[48]. However, we also observed colocalisation of mCherry-caspase-9 and Apaf1-SNAP in HeLa caspase-9KO cells (Supplementary Fig. 5j, k), as well as of 3xFLAG-caspase-9 and Apaf1-SNAP in wild type HeLa cells (Supplementary Fig. 5l). These results show that the foci contain both Apaf1 and caspase-9, two major apoptosome components, corroborating that the Apaf1 foci are a cellular form of the apoptosome.

## Discussion

During mitochondrial apoptosis, activation and oligomerisation of Apaf1 into the apoptosome initiate the caspase cascade which orchestrates cell death[49,50]. Here, we discovered that these events involve accumulation of Apaf1 molecules into multiple foci per cell, consisting of a pleiomorphic cloud-like meshwork of organelle-like dimensions. Formation of Apaf1 foci depends on a specific interaction between the WD40 repeats of Apaf1 and cyt c, described before by the in vitro apoptosome structure[19]. Foci form spontaneously when Apaf1 cannot adopt an inhibited conformation, also in line with the apoptosome in vitro[44,45]. Caspase-9 accumulates in the foci with Apaf1, in line with its recruitment to the apoptosome[10,16,17]. Furthermore, CARD and NOD domains are sufficient for foci formation which is caspase-9 dependent, in accordance with the interactions between Apaf1 and caspase-9 being based on CARD-CARD binding and the NOD driving Apaf1 apoptosome oligomerisation[13,14,23,24]. Thus, while the molecular interactions driving foci formation correspond to those in the apoptosome complex, the overall appearance of the foci is reminiscent of membrane-less compartments[51]. Remarkably, Apaf1 foci can disassemble, which we observed to correlate with cell survival. Collectively, our results suggest that Apaf1 foci are a cellular form of the apoptosome, yet with an irregular ultrastructural organisation and an inherent tendency to dissociate, related to evasion from cell death.

The accumulation of Apaf1 into areas of high density is consistent with activation of caspase-9 requiring its high local concentration[52–54]. For full activity, caspase-9 needs to dimerise and additionally interact with oligomeric Apaf1[13,20,21,42,55,56]. Interestingly, other initiator caspase activation platforms show a similar behaviour. The NLRP3-ASC inflammasome forms upon the oligomerisation of the adaptor protein into a single large cellular speck for activation of caspase-1[57–59].

Caspase-8 filaments oligomerise into the death-inducing signalling complex (DISC), a higher-order oligomeric structure at the plasma membrane, upon accumulation of death receptors such as CD95 and adaptor proteins[60–62]. The association into higher-order oligomeric assemblies of organelle-like dimensions might thus be a general principle for initiating caspase activity cascades.

Such assemblies could spatially constrain activities and thereby facilitate their regulation. Our observation that disassembly of Apaf1 foci correlates with a higher likelihood to evade apoptosis points to a regulatory function. There is growing evidence for cellular mechanisms that facilitate recovery from cell death at various stages of the pathway[63]. For instance, incomplete permeabilisation of the outer mitochondrial membrane can lead to limited caspase activation and thereby cell survival, suggesting the necessity to reach a threshold in caspase activity for completion of apoptosis[64]. In line with this, when we limited effector caspase activity by treating cells with QVD, we observed disassembly of Apaf1 foci and cell survival. It has been suggested that the amount of available procaspase-9 restricts caspase activity to a limited time window for completion of apoptosis[48]. Since caspase-9 is only active when bound to the apoptosome and becomes displaced by new procaspase-9 upon cleavage, its activity would end when procaspase-9 is depleted. This model of a molecular timer[48] is supported by the dynamics of Apaf1 foci. Given that foci contain caspase-9 and that foci formation requires caspase-9, which is expressed as procaspase-9, disassembly of Apaf1 foci could be indicative of depletion of procaspase-9. If this occurred before the threshold of effector caspase activity was reached, cells would survive. This model can also explain our observations of inherent foci disassembly and cell survival upon expression of Apaf1-ΔWD40-GFP. Because this Apaf1 mutant promotes procaspase-9 auto-processing but does not activate effector caspases[13,43,44], depletion of the cytosolic pool of procaspase-9 would occur without eliciting the downstream caspase cascade, resulting in foci disassembly and cell survival. Thus, Apaf1 foci, specifically their disassembly, could underlie a late opportunity for escape from apoptosis.

In our electron tomograms, Apaf1 foci appeared as cloud-like, continuous meshwork structures. The specificity of the molecular interactions required for foci formation indicate a higher-order organisation of the foci. Notably, foci formation depends on caspase-9, which from in vitro data was thought to be recruited to the Apaf1 CARDs once they are in an oligomeric arrangement[10]. This finding, along with our observation that caspase-9 is enriched in the foci together with Apaf1, suggests that caspase-9 has a role in the organisation of Apaf1 foci. Interestingly, in vitro assembled apoptosomes have been observed to cluster in presence of procaspase-9[17]. The irregular meshwork ultrastructure of Apaf1-foci might underpin the instability of the foci. Disassembly and the ability to form repeatedly indicate that the structural arrangement of molecules in the foci is labile and reversible. In line with this, our data support previous reports that binding of cyt c to Apaf1 is transient[27,41]. In contrast to the formation of discrete, stable complexes, a labile assembly could facilitate fast dissociation for ending apoptotic activity. In addition, the irregular, continuous organisation might allow fast bulk sequestration of components without requiring precise stoichiometry.

In summary, our findings suggest that apoptosome function in the cell is based on a dynamic supramolecular organisation of its components. Our study points up structured, large-scale molecular assemblies as a cellular mechanism for spatiotemporal regulation of signalling pathways.

## Methods
### Cloning
See Supplementary Table 1 for oligonucleotide primer sequences. To generate the plasmids for expressing Apaf1-GFP and GFP-Apaf1, the *Homo sapiens* Apaf1-XL[43] cDNA (referred as Apaf1 throughout the text)

was amplified from human mRNA extract (gift from Madeline Lancaster's lab) and introduced into pCI-C-terminal-mEGFP-Gateway and pCI-N-terminal-mEGFP-Gateway destination vectors (gift from Harvey McMahon's lab) using the NheI-HF and KpnI-HF restriction sites or the Gateway recombination sites, respectively. To generate a stable cell line, the Apaf1-mEGFP sequence was introduced into the pLenti6/V5-DEST Gateway vector (Thermo Fisher Scientific, V36820) using the Gateway recombination sites. To generate the plasmid for expressing Apaf1-ΔWD40-mEGFP, the nucleotide sequence corresponding to Apaf1 amino acid residues 1 to 559 was cloned into the pCI-C-terminal-mEGFP-Gateway destination vector using the NheI-HF and KpnI-HF restriction sites. To generate the plasmid for expressing Apaf1-dead-mEGFP, base pairs 2034 to 3812 of the Apaf1-mEGFP sequence were amplified as three separate PCR fragments with primers containing modified codons corresponding to Trp884Asp and Trp1179Asp mutations. The three PCR fragments were assembled by overlapping PCR and introduced into the pCI-Apaf1-mEGFP vector using the EcoRI-HF and KpnI-HF restriction sites. To generate the plasmid for expressing Apaf1-SNAP, the SNAP-tag sequence was amplified by PCR from a gBlock (IDT), digested at Acc65I and Not1 restriction sites and ligated into the pCI-Apaf1-mEGFP vector to replace mEGFP. To generate the plasmid for expressing mCherry-caspase-9, the caspase-9 sequence was amplified by PCR from a gBlock (IDT) and introduced into a pCI-N-terminal-mCherry vector, which corresponds to the pCI-N-terminal-mEGFP-Gateway destination vector in which the mEGFP sequence was replaced by an mCherry sequence. The overlapping PCR fragments were assembled by Gibson Assembly (NEB, E5510). To generate the plasmid for expressing 3xFLAG-caspase-9, a gBlock (IDT) with the sequence of 3x-FLAG linked to the N-terminus of caspase-9 via a short glycine linker (GGGGS) was amplified by PCR and cloned into the pCI-Apaf1-mEGFP vector using the NheI and NotI restriction sites to replace Apaf1-mEGFP. Bacterial strains transformed with all Apaf1-related plasmids were grown at room temperature for 2 days for subsequent purification or cloning procedure because when bacteria were grown at 37 °C, the plasmids could not be recovered. At each cloning step, the desired cDNA sequence was verified by sequencing.

### Generation of the HeLa cell line stably expressing Apaf1-GFP
A HeLa cell line stably expressing Apaf1-GFP was generated by lentiviral transduction. To produce lentiviruses, HEK293T cells were transfected with 6 μg of pLenti6/V5-EXPR-Apaf1-mEGFP mixed with 4 μg of pCMVR8.74 (gift from Didier Trono, Addgene #22036) and 4 μg of pMD2.G (gift from Didier Trono, Addgene #12259) plasmids in Opti-MEM (Gibco, 31985088) using polyethyleneimine (PEI) (1 mg/mL in PBS) transfection reagent at a 1 ng DNA to 4 μL PEI ratio. 48 h later, the medium containing lentivirus was harvested, filtered through a 0.22 μm filter (Elkay, E25PV4550S) and added to HeLa-TetON-WTOTC cells[65] cultured in a media supplemented with 8 μg/mL polybrene (Sigma-Aldrich, H9268). 48 h later, the medium was replaced with fresh medium containing 10 μg/mL of blasticidin (Thermo Fisher Scientific, A1113903) to select the transduced clones. After 10 days, the transduced cells were separated into three non-clonal populations according to their Apaf1-GFP expression level by FACS, based on arbitrary thresholds. All three cell populations showed Apaf1-GFP foci appearance. The population expressing the lowest levels of Apaf1-GFP was designated as the stably expressing cell line hereafter called HeLa-Apaf1-GFP, and was used for subsequent experiments.

### Generation of the HCT116 caspase-9KO cell line
To generate HCT116 caspase-9KO cells, the sgRNA sequence (CGCAGCAGTCCAGAGCACCG) was cloned into LentiCRISPRv2-blasti[66] between Esp3I sites. 293FT cells were co-transfected with 1.86 μg psPAX2 (gift from Didier Trono, Addgene #12260), 1 μg VSVG (gift from Bob Weinberg, Addgene #8454)[67] and 5 μg LentiCRISPRv2-blasti hcaspase-9 using Lipofectamine 2000 (Life Technologies). The

following day the medium was changed to fresh medium containing 20% FBS. Supernatant containing viral particles was harvested at 48 h and 72 h post-transfection and used to infect HCT116 cells. After 24 h of infection, HCT116 cells were allowed to recover in fresh medium without antibiotics for 24 h and then selected with 10 μg/mL blasticidin (Invitrogen) until untransduced cells were killed. Caspase-9 expression in the lysates from the resulting polyclonal cell line was determined by western blotting as described below.

### Cell culture
HeLa cells used in this study were HeLa-TetON-WTOTC[65] cells. All HeLa cells and the U2OS cells were cultured in DMEM/GlutaMAX medium (Gibco, 31966). All HCT116 cells were cultured in McCoy's 5 A/Gluta-MAX medium (Thermo Fisher Scientific, 36600). Both types of media were supplemented with 10% heat-inactivated FBS (Gibco, 10270), 1x MEM-NEAA (Gibco, 11140050) and 10 mM HEPES buffer. HeLa-Apaf1-GFP cells were cultured in a medium additionally supplemented with 10 μg/mL blasticidin (Thermo Fisher Scientific, A1113903) when maintained or stocked for the lab collection. For experiment purposes, a medium without antibiotic was used. All cells were cultured at 37 °C in a 5% humidified $CO_2$ atmosphere. The cell lines were regularly tested for mycoplasma contamination and never tested positive.

### Cellular staining, transfections, and apoptosis induction
To stain mitochondria for all live imaging, room temperature CLEM and cryo-CLEM experiments, cells were incubated with MitoTracker DeepRed (Thermo Fisher Scientific, 22426) at a concentration of 20 nM for 15 min or with MitoSpy Green FM (BioLegend, 424805) at a concentration of 100 nM for 15 min. The cells were washed three times with PBS before fresh medium addition. HeLa cells transfected with the Apaf1-SNAP construct were stained with 1.5 μM SNAP-Cell 647-SiR substrate (NEB, S9102S), according to the manufacturer's instructions. HeLa, HCT116 and U2OS cells were transfected with X-tremeGENE 9 (Roche, 06365787001), FuGENE HD (Promega, E2311) and PEI transfection reagents, respectively, all at a ratio of 1 μg DNA for 3 μL of transfection reagent, in Opti-MEM medium (Gibco, 31985088), according to manufacturers' instructions. The HeLa caspase-9KO cells were transfected using jetPRIME (Polyplus, 101000027) at a DNA to transfection reagent ratio of 1:2 or 1:3, according to the manufacturer's instructions. Whenever used, ABT-737 (Cayman, 11501) was added to the culture medium at a concentration of 10 μM. Whenever used, the caspase inhibitor Q-VD-OPh (APExBIO, A1901), referred to as QVD in the main text, was added at a concentration of 10 μM, except for cells transfected with C3-RFP-hBax plasmid (gift from Richard Youle's lab), where 20 μM of Q-VD-OPh was used. Whenever used, Cisplatin (Sandoz) was added to the culture medium at a concentration of 50 μM.

### Apaf1 knockdown
MISSION predesigned siRNA targeting human Apaf1 (Sigma-Aldrich, PDSIRNA2D - SASI_Hs02_00331274) and MISSION siRNA Universal Negative Control #1 (Sigma-Aldrich, SIC001-1NMOL) were transfected into HeLa cells plated at a confluency of 80% using TransIT-X2 Dynamic Delivery System transfection reagent (Mirus, SKU MIR 6000). 72 h post-transfection, cells were harvested, and proteins were extracted in RIPA buffer supplemented with Halt Protease inhibitor (Thermo Fisher Scientific, 78430). Western blotting was performed as described below for Apaf1 detection, except that the mouse anti-beta-actin (Sigma-Aldrich, A5316) antibody was detected using 1:10000 IRDye 680RD goat anti-mouse secondary antibody (Licor, 926-68070) in 3% BSA supplemented with 0.1% PBS-Tween 20.

### Western blots
For western blots detecting endogenous Apaf1 and tagged Apaf1 constructs, proteins were extracted in RIPA buffer supplemented with Halt Protease inhibitor (Thermo Fisher Scientific, 78430). 10 μg of total

proteins were loaded onto 7% Tricine gels, which ran in cathode buffer (100 mM Tris-base, 100 mM Tricine, 0.1% SDS) and anode buffer (200 mM Tris-HCl, pH = 8.8). The proteins were transferred to a nitrocellulose membrane (Bio-Rad, 1620097) in transfer buffer (192 mM glycine, 25 mM Tris-base, 20% methanol). The membrane was blocked using 5% milk powder in PBS and incubated with primary antibodies (1:1000 rabbit anti-Apaf1 SY22-02 (Invitrogen, MA5-32082) and 1:2000 mouse anti-beta-actin (Sigma-Aldrich, A5316) diluted in 3% BSA in 0.1% PBS-Tween 20) overnight at 4 °C. Secondary antibodies (1:3000 anti-rabbit-HRP (Invitrogen, 65-6120) and 1:3000 anti-mouse-HRP (Dako, P0260), diluted in 3% BSA in 0.1% PBS-Tween 20) were incubated for 1 h at room temperature. Membranes were treated with the Pierce ECL Western Blotting Substrate kit (Thermo Fisher Scientific, 32106) according to the manufacturer instructions and developed using a Western Blot Fusion FX7 Imager (Vilber).

For western blots monitoring caspase-9 and PARP cleavage, the culture medium was collected, the cells were washed with PBS and the wash was collected. The cells were trypsinised, harvested and mixed with both the collected culture medium and the PBS wash to collect all dying cells that detached from the culture plate. Proteins were extracted as for the Apaf1 western blot. 40 μg of total protein were loaded onto NuPAGE 4–12% Bis-Tris gels (Invitrogen, NP0321) and run in MES buffer. Transfer, blocking, antibody treatment and detection were done as for the Apaf1 western blot, except that the primary antibodies were mouse anti-caspase-9 (1:1000) (Cell Signalling, 9508) in 5% milk-PBS-0.1%Tween 20 and mouse anti-PARP antibody (1:500) (BD Biosciences, 556362) in 1% BSA-PBS-0.1%Tween 20. As secondary antibodies, 1:20000 anti-rabbit-HRP (Invitrogen, 65-6120) and 1:20000 anti-mouse-HRP (Dako, P0260) in 5% milk-PBS-0.1%Tween 20 were used.

### Fluorescence microscopy

For all immunofluorescence (except the time-course immunofluorescence), live imaging (except HeLa caspase-9KO and cisplatin induced HeLa Apaf1-GFP live imaging) and microinjection experiments, a Zeiss LSM 710 confocal microscope was used, equipped with a 20x Plan-Apochromat objective with NA = 0.8 and a 63x PlanApo oil-immersion objective with NA = 1.4, operated with the ZEN imaging software (Zeiss). The microscope was equipped with a BiG (binary GaAsP) detector. For live imaging of HeLa caspase-9KO cells and cisplatin induced HeLa Apaf1-GFP, a Zeiss LSM 880 confocal microscope was used, equipped with a 63x oil-immersion objective with NA = 1.4, operated with the ZEN imaging software (Zeiss). The microscope was equipped with a spectral 32 channel GaAsP PMT detector. On both microscopes, lasers at 488 nm, 561 nm, 633 nm were used for green, red and far red fluorophore excitation, respectively. During live cell imaging, cells were kept at 37 °C with a 5% humidified $CO_2$ atmosphere. All confocal fluorescence images (live imaging and immunofluorescence) shown in the figures are single z-planes.

For imaging of the time-course immunofluorescence, of the immunofluorescence on 3xFlag-caspase-9 and of fixed cells expressing mCherry-caspase-9 combined with Apaf1-Snap647, a Nikon Eclipse Ti2 microscope was used, equipped with a 100x CFI Apochromat oil-immersion TIRF objective with NA = 1.49, controlled by the NIS-Elements software (Nikon). A Lumencor SpectraX light source (Chroma) with 470 nm, 555 nm and 640 nm LEDs was used for excitation, with quad band filter set 89000 ET Sedat Quad (Chroma). The emission filter wheel (Nikon) was set to 535 nm, 638 nm and 708 nm for green, red and far-red fluorescence, respectively.

Fluorescence images of resin sections for room-temperature CLEM were acquired using a Ti2 widefield microscope (Nikon) controlled by the NIS-Elements software (Nikon), equipped with a 100x oil-immersion TIRF objective with NA = 1.49, a Niji LED light source (bluebox optics), a NEO sCMOS DC-152Q-C00-FI camera (Andor) and the following filter sets: 49002 ET-GFP (chroma), 49005 ET-DSRed (chroma), 49006 ET-Cy5 (chroma).

### Immunofluorescence experiments

At time points of interest after required treatment, cells plated on coverslips were washed and fixed with 4% paraformaldehyde in PBS, pH 7.2. The coverslips were washed, blocked in 10% goat serum (Sigma-Aldrich, G6767) and 1% saponin (Sigma-Aldrich, 84510) solution and incubated overnight at 4 °C with the primary antibodies. In some experiments, before blocking the coverslips were incubated with 0.3% Triton-X in PBS for 15 min, then blocked in 1% BSA (Sigma-Aldrich, A7906), 0.2% gelatine (Sigma-Aldrich, G7041) and 0.05% saponin (Sigma-Aldrich, 47036) in PBS, and incubated overnight at 4 °C with the primary antibodies. The samples were washed and incubated with secondary antibodies for 1 h at room temperature. The coverslips were washed and mounted with Pro-Long Diamond Antifade Mountant (Invitrogen, P36965) on imaging slides.

Antibodies were used as follows. Primary: Rabbit anti-Apaf1 (Invitrogen, PA5-19893) 1:50, mouse $IgG_{2a}$ anti-Tom20 (Santa Cruz Biotechnology, Sc-17764) 1:200, mouse anti-cytochrome c (BD Pharmingen, 556432) 1:200, and rabbit anti-FLAG (DYDDDDK peptide) (Cell Signalling, 14793S) 1:500. Secondary: Goat anti-rabbit Alexa Fluor 488 (Invitrogen, A11034) 1:200 or 1:500, donkey anti-rabbit Alexa Fluor 488 (Invitrogen, A21206) 1:200, goat anti-mouse Alexa Fluor 647 (Invitrogen, A32728) 1:200, goat anti-mouse IgG2a Alexa Fluor 647 (Invitrogen, A21241) 1:200, goat anti-rabbit Alexa Fluor 568 (Invitrogen, A11011) 1:200, and goat anti-mouse-IgG1 Alexa Fluor 568 (Invitrogen, A21124) 1:200.

### Microinjection of cytochrome c

Microinjections were performed based on a published protocol[39]. Cells were treated with Q-VD-OPh for 30 min to 1 h prior to microinjections. The injection solution contained 10 mg/mL bovine cytochrome c (Sigma-Aldrich, C2037) in 100 mM KCl, 10 mM KPi and 4–8 mg/mL 3000 MW tetramethylrhodamine dextran (Invitrogen, D3308) solution. The control microinjection solution was identical except that it did not contain cytochrome c. Microinjections were performed using a Narishige micromanipulator and an Eppendorf FemtoJet microinjector mounted on a Zeiss LSM 710 confocal microscope (see fluorescence microscopy section) using a 20x objective. Needles (Eppendorf Femtotip, EP5242952008-20EA) were loaded with 1–2 μL microinjection solution. The medium was changed for fresh medium containing Q-VD-OPh after microinjection. The cells were then imaged for at least 4 h using a 63x objective.

For the double microinjection experiment, the cells were imaged for about 10 h before the second microinjection round, which was identical to the first except that the solution contained 0.125 mg/mL 10,000 MW Alexa Fluor 647 dextran (Life Technologies, D22914) instead of tetramethylrhodamine dextran.

### Quantification of cell death by flow cytometry

HeLa cells stably expressing Apaf1-GFP were treated with ABT-737 and after 7 h, 10 h, 16 h, 24 h and 48 h, the culture medium was collected. Cells were scraped and collected with the medium. The cells were washed with FACS buffer (150 mM NaCl, 4 mM KCl, 2.5 mM $CaCl_2$, 1 mM $MgSO_4$, 15 mM HEPES pH 7.2, 2% FCS and 10 mM $NaN_3$) and incubated with Atto633-Annexin V for at least 20 min on ice in the dark. Cells were then washed in FACS buffer and resuspended in 200 μL FACS buffer. Propidium iodide (Sigma-Aldrich, 81845) was added to a final concentration of 2 μg/mL and cells were examined by flow cytometry using a FACSLyric flow cytometer (BD Biosciences). For controls (negative: unstained, positive: stained with Atto633-Annexin V and propidium iodide), HeLa cells stably expressing Apaf1-GFP were treated with 1 μM staurosporine for 2 h to obtain a 1:1 population of apoptotic to healthy cells. For calibration of the FACS machine, staurosporine-induced cells were stained with Atto633-Annexin V or Propidium iodide.

## Correlative microscopy of resin-embedded cells

Correlative microscopy of resin-embedded samples was performed as described before[30,33], with the following modifications: HeLa cells were grown on carbon-coated 3 mm sapphire disks (Engineering Office M. Wohlwend, Art. 500), transfected with Apaf1-GFP and treated with ABT-737 and Q-VD-OPh and stained with MitoTracker DeepRed. High pressure freezing was performed using a Leica HPM100. Freeze-substitution was done with 0.008% (w/v) uranyl acetate in acetone. The resin blocks were cut into 300 to 350 nm (nominal feed) thick sections which were collected onto 200 mesh carbon-coated copper electron microscopy (EM) grids (Agar Scientific, S160). As fiducial markers for correlation, 50 nm TetraSpeck microspheres (Invitrogen, custom order) were diluted 1:200 in PBS and adsorbed for 5 min to the sections.

After fluorescence imaging, 15 nm colloidal gold beads (Agar Scientific) were adsorbed to both sides of the grids. Electron tomography was done on a Tecnai F20 (FEI) operated at 200 kV using a high tilt tomography holder (Fischione, Model 2020). Data was acquired with SerialEM[68]. 2D montages of regions of interest were acquired by transmission electron microscopy (TEM) at approximately 100-130 μm defocus and a pixel size of 1.1 nm. To determine areas of ET acquisition, the montages were correlated to the fluorescence images using TetraSpeck fiducials using MATLAB (MathWorks) scripts[30]. In some cases, MitoTracker DeepRed signals were used for correlation to anchor maps from tomogram acquisition. Dual axis tilt-series[69] were acquired from 60° to −60° with 1° increment in scanning TEM (STEM) mode on an axial bright field detector[70], with a 50 μm C2 aperture at a pixel size of 1.1 nm over a 2048 × 2048 pixel image. Tomographic reconstructions were done using IMOD[71,72]. 3D median filtering was applied for better visibility in the figure panels and movies.

## Vitrification of cells

HeLa WT cells or HeLa cells stably expressing Apaf1-GFP were plated on 200 mesh gold EM grids with a holey carbon film R2/2 (Quantifoil). HeLa WT cells were transfected with Apaf1-SNAP, labelled with SNAP-Cell 647-SiR. All cells were treated with ABT-737 and Q-VD-OPh for around 7 to 8 h and stained with either MitoTracker DeepRed or MitoSpy Green. Grids were vitrified using a manual plunger with a cryostat[73] after manual backside-blotting for 12–15 s using filter paper (Whatman No.1).

## Cryo-fluorescence imaging of vitrified grids

EM grids with vitrified cells were screened for Apaf1 foci by cryo-fluorescence microscopy (cryo-FM) on a Leica EM Cryo CLEM (Leica Microsystems) equipped with a HCX PL APO 50x cryo-objective with NA = 0.9, a DFC9000 GT sCMOS camera (Leica Microsystems), and an EL 6000 light source (Leica Microsystems) in a humidity-controlled room, operated using the LAS X software (Leica Microsystems). 1.5 × 1.5 mm montages of the grids were acquired in the green channel (L5 filter), far red channel (Y5 filter) and bright field channel. At grid squares of interest containing cells with Apaf1 foci, z-stacks were collected. To distinguish between autofluorescent specks in the green channel[74] and Apaf1-GFP signal, the grid squares were additionally imaged in the red channel (N21 filter). Cryo-FM images were used to identify grids with areas of interest for cryo-focused ion beam (FIB) milling.

## Cryo-focused ion beam milling

Thin lamellae of HeLa cells expressing either Apaf1-GFP or Apaf1-SNAP labelled with SNAP-Cell 647-SiR (referred to as SNAP647) were obtained by cryo-FIB milling in an Aquilos 2 Cryo-FIB (Thermo Fisher Scientific) equipped with an integrated fluorescence light microscope (iFLM). Grids were first sputter coated and subjected to GIS coating for 1 min 30 s. Lamellae were prepared semi-automatically with eucentricity, milling angle, stress relief cuts[75], rough, medium and fine milling steps performed automatically using AutoTEM Cryo software (Thermo Fisher Scientific) followed by manual polishing to a target thickness of 200–250 nm. The milling steps with decreasing ion beam current and voltage were similar to those published before[33,76,77]. After rough and fine milling, fluorescence z-stacks of the lamellae were acquired with the iFLM using LED light sources at 470 nm for excitation of GFP and MitoSpy Green, and 625 nm for MitoTracker DeepRed and Apaf1-SNAP647, to assess presence of Apaf1 foci[78,79]. For Apaf1-SNAP647, these z-stacks contained signals of interest and were correlated to overview cryo-EM maps of the lamellae to locate positions for cryo-ET acquisition, using either MATLAB (MathWorks) scripts[30] or ec-CLEM[80]. Apaf1-GFP signals were not detected after fine milling. However, the iFLM images were used as intermediates to correlate the position of the signal in the pre-milling cryo-FM images to the overview cryo-EM maps based on shared image features and thereby determine the position for cryo-ET.

## Cryo-electron tomography

Cryo-ET data of cryo-FIB-milled cells was acquired on a Krios G4 C-FEG cryo-TEM operated at 300 kV equipped with a Falcon 4 detector used in counting mode and a Selectris energy filter, using Tomography 5 Software (Thermo Fisher Scientific). Montage maps of lamellae were acquired at 100 μm defocus at a pixel size of 53.7 Å. These maps were correlated with iFLM images acquired during FIB-milling. At regions of interest, tilt series were acquired using a dose-symmetric tilt scheme[81] using groups of 4 between ±60° at 1° increment and a pixel size of 2.97 Å. A dose of approximately 1 e-/Å² was applied per tilt image. The target defocus was −5 μm. Tilt series were aligned using patch tracking and tomograms reconstructed by SIRT at bin 2 using IMOD[71]. Median filtering was applied for better visibility in the figure panels and movies. Segmentation of the meshwork was done in IMOD[72] using the isosurface function on further median-filtered tomograms. Segmentation models were visualised using UCSF ChimeraX[82].

## Cryo-CLEM data set

Using the pre-milling cryo-CLEM approach (Supplementary Fig. 4b), we targeted 4 Apaf1-GFP signals in vitreous cells, from which we obtained 3 tomograms that contained the meshwork identified as Apaf1 assembly, and 1 tomogram that did not contain the meshwork. The latter is likely due to the position of the lamella in z-direction not matching the z-position of the Apaf1-GFP signal of interest. Inaccurate correlation in z-direction is a disadvantage of pre-milling CLEM.

Using the post-milling cryo-CLEM approach (Supplementary Fig. 4c), we targeted 9 Apaf1-SNAP647 signals on lamellae, from which we obtained 8 tomograms that contained the meshwork identified as Apaf1 foci, and 1 tomogram that did not contain the meshwork. In this one case, the overlay of the fluorescence image and the cryo-EM map of the lamella revealed an imprecise correlation of the MitoSpy Green signal to the mitochondria in the region of interest, indicating that this tomogram might have been mispositioned.

## Fluorescence image analysis

Live cell imaging time points were extracted using the Fiji multi-point marker tool and its multi-point series option[83,84]. To determine the image frames of foci appearance, disappearance, and cell death, we used the following criteria. (1) At least two frames should separate a disappearance from a new appearance event. (2) Occasionally, one or two foci remained visible despite all others having disappeared. In this case, the first frame in which all other foci had disappeared was considered the disappearance frame. (3) Cell death was assigned to the first frame in which a cell shrank, often followed by detachment. (4) When the time point of foci appearance or disappearance could not be unambiguously determined although it had clearly occurred, the event was included in the percentage analysis but not in the time point or lifetime analyses.

For the percentage analysis, cells were counted using the Fiji multi-point marker tool and its multi-point series option[83,84]. For cells dying over the course of imaging, the total number of cells was determined in the first image frame, while the number of cells that

shrank (indicating death) was determined over the entire time of imaging. Since 16% of ABT-737 treated and 26% of ABT-737 and Q-VD-OPh treated cells divided during imaging, the percentage of cells that shrank and thus died is likely overestimated.

For the analysis of the percentages of foci-forming cells at different time points (shown in Fig. 2e) only those four out of six live-imaging experiments analysed in Fig. 2b–d were considered in which more than 30% of cells imaged over the entire imaging session of 18 h showed foci. In the two experiments that were not considered, the percentage of cells with foci was very low at individual time points. Further quantitative analyses were done in R/Rstudio (R Core Team, 2021).

### Fluorescence intensity analysis

The fluorescence intensities of Apaf1-GFP expressing cells that shrank (indicating cell death) with or without foci (Supplementary Fig. 2e) were analysed using a custom-written Python script which can be found here: https://github.com/ibmm-unibe-ch/Apaf1-foci-analysis. The frame prior to foci appearance, or, if no foci appeared, prior to cell shrinkage, was extracted, as well as the frames corresponding to 2 h before these events. The cells were masked and CellProfiler[85] was used to segment the cytosol based on the MitoTracker and Apaf1-GFP signals. The fluorescence intensity of the Apaf1-GFP signal was integrated per cell area based on masks and segmentation. The results were manually curated to remove poorly segmented cells. Analysis was done using GraphPad Prism.

### Quantification of number of foci per cell area

To estimate the number of foci formed in Apaf1-GFP expressing cells surviving or dying following foci formation (Supplementary Fig. 2f), we used single plane images acquired by confocal microscopy. The cells used were treated with ABT-737 only. Foci were manually counted using Fiji multi-point marker tool[83,84] in around 25 randomly selected cells (from a list of cells showing foci, using R) on the frame where Apaf1 foci were best visible. On the same frame, the cell contour was manually drawn, and the delimited area was determined using Fiji. The number of foci per $\mu m^2$ cell area was calculated using GraphPad Prism.

### Statistics

GraphPad Prism was used for descriptive statistics, for statistical tests indicated in the Supplementary Fig. legends, and for preparation of superplots[86]. Median absolute deviations were calculated in Excel.

### Reporting summary

Further information on research design is available in the Nature Portfolio Reporting Summary linked to this article.

## Data availability

Electron tomograms (Fig. 3) are deposited on EMDB[87] with accession numbers EMD-54278, EMD-54279, EMD-54280 and EMD-54281. All other data are contained in the manuscript or available on request from the authors. Source data are provided with this paper.

## Code availability

Custom-written code for analysis of fluorescence intensity is available on GitHub: https://github.com/ibmm-unibe-ch/Apaf1-foci-analysis.

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

## Acknowledgements

We thank Leonardo Almeida-Souza for help with generating the stable Apaf1-GFP cell line, Anja Hagting for help with the microinjection experiments, as well as Antonina Andreeva and Alexey Murzin for help with designing mutant Apaf1. We are grateful to Georg Häcker, Madeline Lancaster, Richard Youle and Harvey McMahon for sharing cell lines, cDNA and plasmids. We thank Sean Munro for helpful discussions and support. We thank the electron and light microscopy facilities of the MRC-LMB as well as the Microscopy Imaging Centre (MIC) of the University of Bern and the Dubochet Centre for Imaging (DCI Bern) for microscope access and support with data collection. We thank the Flow Cytometry facility of the MRC-LMB for help with cell sorting. We thank Ori Avinoam for critical reading of the manuscript and members of the Kukulski group for helpful discussions. Work in the group of W.K. was supported by the Medical Research Council, as part of United Kingdom Research and Innovation (also known as UK Research and Innovation) under award MC_UP_1201/8, by the University of Bern, the NCCR TransCure a National Centre of Competence in Research of the Swiss National Science Foundation (SNSF) (185544), the SNSF project 201158 and the Novartis Foundation for Medical-Biological Research. Work in the group of T.K. was supported by the SNSF project 201199. Work in the group of S.W.G.T. was supported by the Cancer Research UK (DRCNPG-Jun22\100011 and A20145). Acquisition of the Aquilos 2 was supported by the SNSF R'Equip 198524. For the purpose of open access, the authors have applied a CC BY public copyright licence to any Author Accepted Manuscript version arising.

## Author contributions

A.C.B., I.G., C.K., A.S., D.R.-K., L.W., and J.S.R. carried out experiments and analysed data. T.K., T.L., and S.W.G.T. contributed to experiment planning and data analysis, and supervised parts of the work. W.K. supervised the project and analysed data. A.C.B. and W.K. conceived the study, planned experiments, and wrote the manuscript, with input from all authors.

## Competing interests

The authors declare no competing interests.
