## [Transparent Peer Review file · Nature Communications]

Large transient assemblies of Apaf1 constitute the apoptosome in cells

Corresponding Author: Professor Wanda Kukulski

Version 0:

Reviewer comments:

Reviewer #2

(Remarks to the Author)

The authors have dealt with many of the points raised in my previous comments, but have failed to address several important points.

1. The authors have declined to image endogenous caspase-9 to determine whether this behaves similarly to Apaf-1 in terms of forming transient assemblies (Point 1). I'm a bit surprised by this as I feel that this is a very reasonable request and should be technically feasible with a good antibody, but the authors have declined to do this. The lack of information concerning caspase-9 behaviour relative to Apaf-1 should be acknowledged as a significant limitation of the study.
2. The authors have also declined to address the molecular timer model proposed by Bratton and colleagues regarding caspase-9 disengagement from Apaf-1 (EMBO J) which would have required a simple experiment involving overexpression of a processing-deficient mutant of caspase-9 (point 3). The processing of caspase-9 and its disengagement from Apaf-1 may impact on the transient nature of the assemblies reported. I felt that this was in the scope of the study, but the authors clearly disagree.
3. With regard to the overall signal brightness dropping in Figure 2A at the 12h 35 min timepoint (point 6), the authors present arguments that are not that convincing to me. The brightness of the 12h 35min image is manifestly much less than that of the other two images. I would prefer that the authors replace this image series with a series of images where there is more convincing disappearance of the Apaf-1 spots, where signal bleaching is much less of a factor. As it stands, the claim that these assemblies are transient in nature could be an artifact due to sample bleaching.

Reviewer #3

(Remarks to the Author)

I had two specific questions for the authors, which they chose to dismiss politely. They argued that my first question lacked support from the evidence they presented. However, I could clearly observe proximity between the foci and mitochondria. Without formal quantification (which I had requested), either interpretation remains plausible.

My second question sought to clarify whether a specific threshold of cytochrome c was necessary to trigger foci formation. Given that cytochrome c release from mitochondria is not all-or-nothing—and considering earlier studies by Zhivotovsky showing that different cell types require varying concentrations of microinjected cytochrome c to induce cell death (likely due to differences in endogenous caspase inhibitor levels)—I felt this was an important, nontrivial question.

In conclusion, the manuscript remains unchanged. I was supportive in the first round of revisions, and I remain positive now, despite my questions remaining unresolved.

Luca Scorrano

Version 1:

Reviewer comments:

Reviewer #2

(Remarks to the Author)

The authors have now addressed my comments satisfactorily.

Reviewer #3

(Remarks to the Author)

Thank you for addressing my comments in the framework of how instructed by the Editors. I have no further concerns given that you did exactly as told.

Luca Scorrano

Point-by-point response to the Reviewers comments

Reviewer #2 (Remarks to the Author):

The authors have dealt with many of the points raised in my previous comments, but have failed to address several important points.

We thank the Reviewer for assessing our revised manuscript. For the previous revision, we were editorially advised to only address technical issues, which we did. We assumed that the Reviewers were aware of this editorial decision, and we therefore kept many of our responses to comments that went beyond technical issues short. We apologise if this resulted in our responses not being satisfying.

1. The authors have declined to image endogenous caspase-9 to determine whether this behaves similarly to Apaf-1 in terms of forming transient assemblies (Point 1). I'm a bit surprised by this as I feel that this is a very reasonable request and should be technically feasible with a good antibody, but the authors have declined to do this. The lack of information concerning caspase-9 behaviour relative to Apaf-1 should be acknowledged as a significant limitation of the study.

We agree that this is a reasonable request and have therefore now determined whether caspase-9 also accumulates in the Apaf1 foci. Our results show that indeed, caspase-9 accumulates in Apaf1 foci, and we show these results in Fig. 5j and Supplementary Fig. 5j-l.

Our experimental setup is however different than suggested by the Reviewer, and we would like to explain the rationale behind these adjustments.

As the Reviewer notes, addressing this question by immunofluorescence (IF) for detecting endogenous caspase-9 would be ideal but requires a good antibody. Many antibodies against caspases-9 are commercially available, and we have used some of them successfully for western blots (Supplementary Fig. 5h and i). We thus tried four different antibodies for IF, as suggested by the Reviewer. Unfortunately, in our hands, none of the tested antibodies was specific enough in IF to unambiguously detect endogenous caspase-9.

We thus next attempted to express tagged caspase-9 constructs. However, high expression levels of caspase-9 lead to an accelerated progression of cell death events (Malladi *et al.*, EMBO J. 2009; see also point 2), which we also observed in our HeLa cells, even in presence of QVD which inhibits events downstream of caspase-9. We therefore turned to caspase-9KO HeLa cells, reasoning that expressing caspase-9 constructs in these cells would result in lower total levels of caspase-9 than in wild type cells.

We co-transfected caspase-9KO HeLa cells with Apaf1-Snap and either mCherry-caspase-9 or 3xFLAG-caspase-9, which we detected by IF using an anti-Flag antibody. In both these settings, we were able to observe foci that contained both Apaf1 and caspase-9 accumulations. These results show **that Apaf1 foci also contain caspase-9, supporting our main conclusion that the foci correspond to the apoptosome.** We now show these results

in Fig. 5j, Supplementary Fig. 5j-l, and refer to them in the Results section of the main text as follows:

Since caspase-9 is required for formation of Apaf1 foci, we asked if the foci contain caspase-9. We therefore co-expressed a FLAG-tagged caspase-9 construct and Apaf1-SNAP in HeLa caspase-9KO cells treated with ABT-737 and QVD. By immunofluorescence of these cells, we found that caspase-9 colocalised with Apaf1 in foci (Fig. 5j). Foci were rare to observe when caspase-9 and Apaf1 constructs were co-expressed (Supplementary Fig. 5j), in line with an acceleration of apoptotic events upon overexpression of caspase-9⁴⁸. However, we also observed colocalization of mCherry-caspase-9 and Apaf1-SNAP in HeLa caspase-9KO cells (Supplementary Fig. 5j and k), as well as of 3xFLAG-caspase-9 and Apaf1-SNAP in wild type cells (Supplementary Fig. 5l). These results show that the foci contain both Apaf1 and caspase-9, two major apoptosome components, corroborating that the Apaf1 foci are a cellular form of the apoptosome.

In the Discussion section:

Caspase-9 accumulates in the foci with Apaf1, in line with its recruitment to the apoptosome^{10, 16, 17}.

and:

This finding, along with our observation that caspase-9 is enriched in the foci together with Apaf1, suggests that caspase-9 has a role in the organisation of Apaf1 foci.

While we observed colocalization of Apaf1 and caspase-9 repeatedly and with both mCherry and FLAG detection, foci were rare in these conditions, and we had to search the coverslips for cells with foci. This made time-lapse live imaging difficult, which relies on a limited number of randomly selected regions of interest to display foci at some point during an imaging session. We therefore imaged fixed cells, allowing a dedicated search and counting of cells with foci at a given time point and in multiple experimental repeats.

Having found conditions to detect colocalisation in caspase-9KO cells, we next also co-transfected wild type HeLa cells with Apaf1-Snap and 3xFlag-caspase-9. Although also rare, as expected, we succeeded to observe **cells with foci that contained both Apaf1 and caspase-9 in wild type HeLa cells**; an example being shown in Supplementary Fig 5k.

There may be multiple possible reasons for foci rareness under conditions of overexpression of caspase-9. One is probably, as mentioned before, the acceleration of events towards cell death (Malladi *et al.*, EMBO J 2008) which is likely to specifically affect events upstream of QVD inhibition. Secondly, the ratio of expression of the two components (caspase-9 and Apaf1) could play a role in foci formation, if the two components must be present at a specific ratio. This is in line with our other observation that foci formation depends on availability of caspase-9.

2. The authors have also declined to address the molecular timer model proposed by Bratton and colleagues regarding caspase-9 disengagement from Apaf-1 (EMBO J) which would have required a simple experiment involving overexpression of a processing-deficient mutant of caspase-9 (point 3). The processing of caspase-9 and its disengagement from Apaf-1 may impact on the transient nature of the assemblies reported. I felt that this was in the scope of the study, but the authors clearly disagree.

We did not address this question in the first revision as we considered it not required for our conclusions, and we were editorially advised that further biological insights were not required, as explained above.

This being said, we have tried to see if the processing-deficient mutant of caspase-9 (caspase-9TM) would alter the lifetime of the foci, as might be expected by the molecular timer model. The experiment, however, turned out to be not trivial, because overexpression of caspase-9 constructs accelerates cell death (see also response to point 1), and caspase-9-TM does so at even lower expression levels than wild type caspase-9 (Malladi *et al.*, EMBO J. 2009). This makes it difficult to capture transient events such as Apaf1 foci, and we were therefore so far unable to image sufficient numbers of cells with foci under these conditions.

During our attempts, western blots to test our caspase-9-TM construct in caspase-9KO HeLa cells showed that it is active in transmitting downstream signal, as PARP cleavage does occur (see figure below, left blot). This is in line with previous reports that preventing caspase-9 cleavage does not prevent downstream cleavage events (Srinivasula *et al.*, Nature 2001) and that in fact activity is enhanced for caspase-9-TM (Malladi *et al.*, EMBO J 2009).

Nevertheless, caspase-9-TM is expected to have a different affinity to Apaf1 than caspase-9 and could thus prolong foci lifetime, based on the molecular timer model. But also, according to the model, the available amount of uncleaved caspase-9 would influence foci lifetime, making the expression levels of caspase-9 constructs highly relevant to the experimental setup – as shown in Malladi *et al.*, EMBO J 2009. Our western blot on the right (Figure below) indicates that our WT caspase-9 construct is present at much higher protein levels than caspase-9-TM. Thus, disentangling these different effects would require adjusting expression levels of the different caspase-9 constructs very carefully in different cell lines. While possible and very interesting, we hope that the Reviewer agrees that our preliminary results show that such experiments would require a significant investment of time that goes beyond the scope of this revision. Most importantly though, we are convinced that our current conclusions do not depend on further validating the molecular timer model.

3. With regard to the overall signal brightness dropping in Figure 2A at the 12h 35 min timepoint (point 6), the authors present arguments that are not that convincing to me. The brightness of the 12h 35min image is manifestly much less than that of the other two images. I would prefer that the authors replace this image series with a series of images where there is more convincing disappearance of the Apaf-1 spots, where signal bleaching is much less of a factor. As it stands, the claim that these assemblies are transient in nature could be an artifact due to sample bleaching.

We have now replaced the example in Figure 2a by a different image series.

The analysis we presented in our previous response to the Reviewer intended to show that throughout our data set, there is indeed some photo bleaching in the relevant time span, but it is too little to affect the detection of Apaf1 foci. This is the case because the degree of bleaching is less than the cell-to-cell variability when foci are present and detectable.

We would additionally like to emphasize that bleaching affects all cells in the field of view similarly. If disappearance of foci in cell A were an artifact of bleaching, a neighbouring cell B, having similar total fluorescence levels at the start of the imaging, would not be expected to show foci after foci had disappeared in cell A, as the fluorescence in cell B would have been bleached similarly to that in cell A, hence below detection levels of foci. However, we observed multiple instances where neighbouring cells of similar initial fluorescence levels show foci after each other. One such example is shown below. The two frames are separated by 105 min, corresponding to 7 acquired image frames. The cells that show foci in frame B do so after foci that appeared in frame A have disappeared. The cytosolic levels of the three involved cells are similar within each frame.

We would further like to point out that the disappearance of foci is also reflected in our IF results, which are not based on time-lapse imaging and thus not subject to bleaching over time (Fig 2g). The number of cells with foci of endogenous Apaf1, fixed at different time points, follows the same distribution as in live cell imaging (Fig. 2e). For IF, each time point is an individual sample, imaged once; hence not affected by photo bleaching. The fact that the number of cells with foci decreases after 12-14 hours after induction of apoptosis both in IF and live cells, supports that the disappearance of foci over time is an intrinsic phenomenon and not a bleaching artefact.

We do not exclude, however, that the protein levels of Apaf1 within individual cells change over the lifetime of foci (e.g., due to protein degradation), which could possibly contribute to foci disassembly, and would be reflected in changes in the overall fluorescence intensity of cells.

Reviewer #3 (Remarks to the Author):

I had two specific questions for the authors, which they chose to dismiss politely. They argued that my first question lacked support from the evidence they presented. However, I could clearly observe proximity between the foci and mitochondria. Without formal quantification (which I had requested), either interpretation remains plausible.

We thank the Reviewer for their continued support of our study. As mentioned in the response to Reviewer #2, we were editorially advised to only address technical issues in the previous revision. We assumed the Reviewers were aware of this, but we apologise if this was not clear.

This being said, we do not think that there is a striking proximity between sites of Apaf1 foci formation and mitochondria that would call for the analysis that the Reviewer suggests. The frames from time-lapse movies shown in the Figure below illustrate why we think so. In all three examples, the image shown is a maximum projection of the first time point in which foci are unambiguously detectable in the indicated cell, thus these images are representative for *where the foci start*, which is what the Reviewer asked in the first instance. The yellow arrowheads indicate some of the foci that are in a substantial distance from the next mitochondria. While indeed there are foci in areas crowded with mitochondria, many foci form in areas devoid of mitochondria. The cytoplasm is generally crowded with mitochondria and there are many Apaf1 foci, thus at the limited resolution of confocal microscopy proximity inevitably appears. We thus agree with the Reviewer that we cannot entirely exclude the

possibility that foci do form in proximity of mitochondria, at least to some extent. However, given the observation of many foci forming far away from mitochondria, this would be a subtle and thus difficult to quantify effect, requiring additional approaches to test its relevance. Furthermore, it would require comparison to a control structure that is similarly frequent as foci but not associated with mitochondria.

Taken together, we think that the question is difficult to unambiguously answer, and we do not think that there are sufficiently obvious indications from our data to make such an analysis pertinent. Most importantly, we do not think that any of the conclusions in our paper would be affected by obtaining such a result. For these reasons, we did not further address this question in this study.

My second question sought to clarify whether a specific threshold of cytochrome c was necessary to trigger foci formation. Given that cytochrome c release from mitochondria is not all-or-nothing—and considering earlier studies by Zhivotovsky showing that different cell types require varying concentrations of microinjected cytochrome c to induce cell death (likely due to differences in endogenous caspase inhibitor levels)—I felt this was an important, nontrivial question.

We agree with the Reviewer that this is a nontrivial and important question. It is, however, also not trivial to experimentally address, because we found the microinjection system to be rather imprecise for controlling the exact volume injected into individual cells. The mentioned study by the Zhivotovsky lab has inspired and guided our microinjection experiments. We thus appreciate this reference for varying cytochrome c concentration microinjected into cells. However, in our hands, it has been difficult to accurately control the microinjected volume. We thus feel that addressing this question would require establishing a different experimental

system, for example using cantilevers with a microfluidics system, as used for microinjection of bacteria into HeLa cells (Gäbelein *et al.*, ACS Synthetic Biology, 2022). In addition to the injection volume, the volume of the HeLa cell will determine the final cytosolic concentration of cytochrome c. The HeLa total cell volume can probably vary by a factor of 5 from cell to cell <https://bionumbers.hms.harvard.edu/bionumber.aspx?&id=103725&ver=20>

This further complicates the estimation of the threshold cytochrome c concentration.

Apart from the experimental difficulties, we do not think that determining the threshold concentration of cyt c for Apaf1 foci formation is required to support any of our current conclusions, and we therefore think these experiments would go beyond the scope of this study. Nevertheless, we agree with the Reviewer that this would be an interesting question to explore in the future.

In conclusion, the manuscript remains unchanged. I was supportive in the first round of revisions, and I remain positive now, despite my questions remaining unresolved.

Luca Scorrano

Additional changes to the manuscript:

For the quantifications of our experimental data, we now report medians instead of means. Many of our analyses are based on either small N (such as percentages of cells, when each N is an entire experiment) or are not normally distributed (such as lifetimes of foci per cell). We realised that it is therefore more appropriate to report medians instead of means. We have modified all figure plots accordingly, and adjusted the numbers given throughout the manuscript including the figure legends.

We have also removed statistical tests in those instances when the compared data sets consist of $N < 5$ (e.g. when the compared data are percentages of cells per experiment and N is the number of experiments), as we realised that in such cases the results of the tests may not be meaningful, and it is more appropriate to not use them.